# Influence of the COVID-19 Pandemic on Musculoskeletal Complaints and Psychological Well-Being of Employees in Public Services—A Cohort Study

**DOI:** 10.3390/jpm13101478

**Published:** 2023-10-09

**Authors:** Leonie Wolf, Philipp Maier, Peter Deibert, Hagen Schmal, Eva Johanna Kubosch

**Affiliations:** 1Department of Orthopedics and Trauma Surgery, Faculty of Medicine, University Hospital, Albert-Ludwigs-University of Freiburg, Hugstetter Strasse 55, 79106 Freiburg, Germany; leonie.wolf@uniklinik-freiburg.de (L.W.); hagen.schmal@uniklinik-freiburg.de (H.S.); 2Institute for Exercise- and Occupational Medicine, Faculty of Medicine, University Hospital, Albert-Ludwigs-University of Freiburg, Hugstetter Strasse 55, 79106 Freiburg, Germany; philipp.maier.ibam@uniklinik-freiburg.de (P.M.); peter.deibert@uniklinik-freiburg.de (P.D.)

**Keywords:** COVID-19, musculoskeletal pain, mental well-being, quality of life, working from home, healthcare workers

## Abstract

Background: The COVID-19 pandemic resulted in unprecedented restrictions on public and private life. The aim of the study was to investigate the impact of the COVID-19 pandemic on the physical and mental health of employees in the public sector, especially patient-related professions. Methods: For the data collection in summer 2021, an online questionnaire was used. Compared to a pre-pandemic point of time, the prevalence, frequency, and intensity of musculoskeletal pain, mental well-being, health status, and quality of life were recorded. Results: The questionnaire was completed by 1678 employees (f: 1045, m: 617). A total of 1504 employees (89.6%) were affected by complaints. Compared to before the pandemic, the prevalence and intensity of musculoskeletal complaints and psychological distress increased significantly. Patient-related professions (*n* = 204) showed significantly higher levels of stress and discomfort in several aspects (*p* < 0.05). Conclusions: Due to the COVID-19 pandemic, there was an increase in musculoskeletal complaints and a decrease in psychological well-being. Preventive factors related to mental health were identified as male gender, a middle- to older-age group, employees with children, and working from home. Attention should be drawn to these findings and prevention strategies should be brought into focus to strengthen the employees’ health. Special focus should be drawn to patient-related professions who are particularly confronted with pandemic-associated challenges.

## 1. Introduction

After the COVID-19 pandemic was declared by the World Health Organization (WHO) on 11 March 2020, more than 750 million people worldwide have been confirmed to be infected with COVID-19 [1,2]. To contain the pandemic, infection control measures were taken, resulting in unprecedented restrictions on everyday life.

### 1.1. Course of the COVID-19 Pandemic in Germany

While only sporadic cases occurred in Germany until the end of February 2020, the first COVID-19 wave began in early March [3]. In order to contain the pandemic, Ordinances of the State Government on Infection Control Measures against the Spread of the SARS-CoV-2 Virus (Corona Ordinance—Corona VO) were adopted at the state level to implement federal requirements and were continuously adapted to the infection event [4]. These included the closing of educational institutions, recreational facilities, and drastic contact restrictions. Consequently, social isolation increased, and homeschooling and childcare by, largely, working parents became necessary. Therefore, people were required to work from home, which, due to the blurring of work and private life, let to a double burden for many.

### 1.2. Musculoskeletal Complaints and Mental Well-Being

Musculoskeletal complaints not only reduce the functionality and quality of life of those who are affected [5]; with a share of 21.5%, musculoskeletal complaints were also the leading cause for incapacity of work in 2016 [6]. In the survey “Work-related musculoskeletal disorders: prevalence, costs and demographics in the EU”, conducted by the European Agency for Safety and Health at Work in 2015, 37% of employees reported a negative impact of their work on their health. Among work-related musculoskeletal complaints, back pain and discomfort in the shoulder and neck region were described more frequently than pain in the lower extremity [7].

Regarding work life, various studies have investigated how musculoskeletal complaints are influenced by workplace design. It was found that musculoskeletal complaints are raised mainly by employees who have inadequate workplace equipment [8,9,10,11]. Psychosocial work style factors include aspects such as social interactions, lack of breaks, high work stress or pressure, and physical inactivity [10,11,12]. Additionally, psychosocial working conditions also have an impact on mental health. In 2013, the 12-month prevalence for mental disorders in Germany was 27.7% in total [13]. The most common diagnoses were anxiety disorders with 15.4%, followed by affective disorders with 9.8%, which also includes depression [13]. The consequences of mental illnesses are both psychosocial, e.g., in the form of reduced participation in everyday activities, and occupational, e.g., in the form of incapacity to work or early retirement [14]. In 2021, with 12%, mental stress represented the second-largest share of days of incapacity to work [6].

### 1.3. Patient-Related Professions

Early in the pandemic, patient-related professions (PRP) were considered to be the most valuable resource, and their well-being and safety were maintained through adequate protective measures and psychological support [15]. On a daily basis, they were exposed to both the risks and consequences of the pandemic through their work. Wei et al. showed up to a four-times higher rate of infection among healthcare workers than non-healthcare workers [16]. The risk of transmission to family members, the feeling of social isolation, and changes at work are seen as other stressors [17].

### 1.4. Aim of the Study

Health and quality of life are defined by physical and mental well-being, social environment, living and working conditions, and many more. Through regulations, numerous changes on these parameters took place during the pandemic [18,19].

The aim of the present study was to investigate the impact of the COVID-19 pandemic on physical and mental health compared to a pre-pandemic point of time in employees in public services. Particular attention was paid to the group of patient-related professions (PRP).

## 2. Materials and Methods

This cohort study targeted public service employees in Germany and took place between early May and mid-October 2021. Inclusion criteria were both an age above 18 (years) and an employment at the university hospital, the regional council, the district administration, or other public services. No exclusion criteria were defined. The study was approved by the Ethics Committee of the Albert-Ludwigs-Universität (nb. 21-1123) and complies with the tenets of the Declaration of Helsinki.

### 2.1. Questionnaire

To survey how the COVID-19 pandemic influenced the health of employees, a retrospective, anonymized online survey was created by using REDCap^®^ (version 10, Research Electronic Data Capture, Vanderbilt University). Socio-demographic data such as gender, age, occupational group, and family circumstances were collected.

The work situation was surveyed by questions about the location of the workplace, satisfaction with the work situation under pandemic conditions, new challenges, and work-related and private stressors. If the employees worked at least partially from home at the time of the survey, further questions opened up about the work situation at home.

For the assessment of physical and psychological complaints, the areas “shoulder and neck”, “back”, “lower extremity”, “headache or burning eyes”, “general exhaustion”, and “sleep problems” were surveyed. The prevalence, frequency, intensity of pain, and whether medication was taken were recorded. The information at the time of the survey was related to the previous year, a point of time before the pandemic, either by the scale levels or by further questions. Changes in health status and quality of life compared with the pre-pandemic period could be indicated using a 7-point Likert scale (“−3 = significantly worse” to “3 = significantly better”).

Psychological distress was evaluated by the validated Patient Health Questionnaire 4 (PHQ-4), an ultra-brief screening scale for anxiety and depression [20]. The PHQ-4 is composed of the Patient Health Questionnaire-2 (PHQ-2), a screening tool for depression, and the Generalized Anxiety Disorder Scale (GAD-2), a screening tool for anxiety, consisting of two items each [21]. For each item, 0 to 3 points are scored, depending on the frequency of the symptoms. Scores between 0–2 points are rated as normal, 3–5 points as mild, 6–8 points as moderate, and 9–12 points as severe psychological distress. Scores of ≥3 points on the PHQ-2 or GAD-2 are classified as conspicuous screening results for depression or anxiety, respectively.

The online questionnaire was made available to employees on the intranet of the respective institutions.

### 2.2. Analysis

SPSS Statistics^®^ (version 28.0, SPSS Inc., Chicago, IL, USA) was used for statistical analysis. The descriptive evaluation of the frequencies was given both absolutely and relatively in percent. Depending on the scale level and the research question, the mean and median were also calculated.

In order to determine a possible significant difference between the groups or variables, the chi-square test was used, whereby the presupposed minimum frequency was not below five. To test whether significant differences within the same group but at varying time periods occurred, the paired-samples *t*-test was used. Results with a probability of error of “*p* < 0.05” were classified as statistically significant.

## 3. Results

Our cohort study included data from 1678 participants (*n* = 1678). Eight questionnaires were rejected due to almost complete missing data.

### 3.1. Socio-Demographic Characteristics

With a share of 642 employees (38.3%), the university hospital was the most represented, followed by the regional council with 389 (23.2%), the district administration with 289 (17.2%), and an energy supplying company with 316 (18.8%). A relief organization formed the smallest share with 42 participants (2.5%).

Further details on the socio-demographic characteristics are shown in Table 1.

### 3.2. Work Situation during the COVID-19 Pandemic

At the time of the survey, 873 of the participants (52%) worked at their usual place in the office. Due to the pandemic, 628 employees (37.4%) worked as much as possible from home, and 121 of the employees (7.2%) worked exclusively from home. Further data on the work situation during the COVID-19 pandemic are shown in Table 2.

With regard to the work situation under pandemic conditions, significant higher values (*p* < 0.001) for dissatisfaction, more challenges, and work-related and private stress were found for employees from the PRP (medical staff and nurses) compared to the other professional groups.

In addition, employees with children were found to be significantly more satisfied with the work situation under pandemic conditions than employees without children (*p* < 0.01). This difference did not exist before the COVID-19 pandemic.

Female employees and employees from the “45–64 years” age group experienced significantly more challenges at work than male employees or employees from younger age groups (*p* < 0.05). While work-related stress increased significantly among employees in the “45–65 years” age group (*p* < 0.002), private stress increased significantly among female employees, employees from the “30–44 years” middle aged group, and employees with children (*p* < 0.05 and *p* < 0.01, respectively).

Employees who worked as much as possible or exclusively from home due to the pandemic were significantly more satisfied with the work situation at the time of the survey than employees with an office workplace (*p* < 0.001). Furthermore, employees with an office workplace showed a significantly higher increase in work-related and private stress (*p* < 0.001).

### 3.3. (Musculoskeletal) Complaints and Mental Well-Being

Physical and psychological complaints that existed within the last two weeks at the time of the survey are presented in Table 3. A total of 1504 participants (89.6%) were affected by (musculoskeletal) complaints. Compared to the previous year, a point of time before the onset of the pandemic, a significantly higher prevalence was found in all categories (*p* < 0.001). Patient-related professions reached a significantly higher prevalence and an increase of the intensity of complaints in the shoulder and neck region, backpain, headache or burning eyes, and the feeling of general fatigue (*p* < 0.05).

There was an increase in the number of employees taking medication for their health conditions compared to the pre-pandemic situation. Relative to the group of employees with complaints, the number of medications taken decreased, except for the shoulder and neck areas and sleeping problems. Pain medication for musculoskeletal complaints was the most common, with relative use ranging from 17.7–25.5%. Significantly higher use of pain medications was observed in patient-related professions (*p* < 0.001). Medications for the feeling of general fatigue or sleeping problems were taken by 5% and 7.2% of the sufferers, respectively.

Moreover, female employees showed significantly higher prevalences for physical and psychological complaints, except for lower extremity pain, than male employees (*p* < 0.001). Employees with an office workplace also showed higher prevalences in all areas in comparison to employees who predominantly worked from home (*p* < 0.05).

In addition, significant differences could be found between the age groups. While the prevalence of lower extremity pain and the prevalence of sleeping problems were higher in the oldest age group “45–64 years” (*p* < 0.001), the middle- and younger-age groups were more frequently affected by headaches and burning eyes (*p* < 0.001).

The PHQ-4 showed mild-to-severe psychological distress in 823 employees (49.0%). Broken down into the subscales, 15.4% and 21.2% of employees showed conspicuous screening results for anxiety and depression. Significantly higher scores were reached by patient-related professions in regard to psychological distress and the subscale for depression was 28.9% (*p* < 0.05), as shown in Table 4.

Significantly higher PHQ-2 values were found for the subgroups of female employees, employees from the age group “18–29 years”, and employees without children (*p* < 0.05).

Employees with an office workplace showed higher PHQ-2 and GAD-2 values (depression (*p* < 0.001), anxiety (*p* < 0.05)) and higher PHQ-4 values (*p* < 0.001) compared to employees who work predominantly from home.

Furthermore, it was shown that the higher the psychological distress of the employees, the higher their dissatisfaction with the current work situation was. They showed a greater increase in private and work-related stress, higher prevalences of (musculoskeletal) complaints, and a significant decrease in health status and quality of life (*p* < 0.001).

## 4. Discussion

The present study showed an increase in musculoskeletal complaints and mental health strains among public service employees, especially among patient-related professions, during the COVID-19 pandemic.

### 4.1. Work Situation during the COVID-19 Pandemic

Being less or not at all satisfied with the current work situation was stated by 28.3% of the employees. In addition, the majority reported increased work-related and private stress, and more challenges at work. Comparable results were achieved by the survey of the Federal Ministry of Labor and Social Affairs, titled “Work Situation and Stress Perception in the Context of the COVID-19 Pandemic in September 2022 […]”. In this survey, 28% of the responders were less satisfied with their work situation and 40% reported an increase in stress levels [22]. Several studies and reviews examined work-related stressors. These included job and future uncertainties, financial worries (especially unemployment or job changes), and long periods of isolation, for example, due to limited contact with coworkers [23,24,25]. Respondents from the present study experienced the increased workload and the general work situation as work-related stressors, primarily due to a lack of contact with colleagues and insufficient hygiene measures.

4.2. (Musculoskeletal) Disorders and Mental Well-Being

Considering the present results, it is remarkable that people reacted differently to the pandemic. Although the majority reported a worsening of their health status and quality of life, few employees also experienced improvement. The improvement in health status was primarily attributed to a reduction in the incidence of seasonal infections through infection control measures. In the Robert Koch Institut’s (RKI) longitudinal study of quality of life during the pandemic, it became apparent that the quality of life declined significantly as the duration of the pandemic and the incidence of infection and the associated tightening of prescriptions increased. Divided into subscales, social and psychological quality of life were more affected than physical and environmental quality of life [26].

During the COVID-19 pandemic, the prevalence and intensity of physical and psychological complaints increased significantly. The shoulder, neck, and back region were most affected; more than half of the employees also reported headaches or burning eyes. One possible approach to the rationale for increased musculoskeletal complaints is physical activity. Due to the restrictions of the COVID-19 regulations, the practice of physical activity in the context of recreational facilities or sports facilities was not possible in the usual way [27,28,29]. At the same time, physical activity is widely known as a preventive measure for health. Initial meta-analyses already showed a possible effect of physical activity as a resource for maintaining health status during the pandemic [27,29].

According to the vulnerability-stress coping model, a mental disorder manifests when protective factors for coping with stress are insufficient and vulnerability increases [30,31,32]. Brakemeier et al. described the COVID-19 pandemic as a “unique multidimensional and potentially toxic mental health stressor”. This stressor is characterized by unpredictable duration, subjectively experienced loss of control, individual and systemic impacts, and limited access to protective factors and support systems [33]. Related to mental well-being, about half of the respondents showed mild-to-severe psychological distress in PHQ-4. Meta-analyses and systematic reviews consistently showed psychological distress during the pandemic, with associated higher rates of depression, anxiety, and sleeping problems [34,35,36,37]. Present results showed a significant increase in the general feeling of exhaustion, a symptom of psychological distress. In addition, it was found that the greater the psychological distress of employees were in the PHQ-4, the more significant the increase in the prevalence of (musculoskeletal) complaints. Moderate-to-severe levels of anxiety and depression were seen in our study, which is in line with results from the survey by the RKI in Summer 2021, using the same measurement tools. After initial resilience, the RKI was able to show an increase in the values with the duration of the pandemic [35]. There was also a significant increase in the prevalence of sleeping problems, affecting 43% of the employees in our study within the last two weeks. Jahrami et al. determined similar results in a meta-analysis, with a global prevalence of 40.5% for sleeping problems during the COVID-19 pandemic, a prevalence of 42.5% of which was found with existing lockdown [38].

In addition to work-related stress, private stress also increased for almost half of the employees. Participants cited homeschooling or childcare as the leading stressor in their personal lives. Further stressors were the lack of recreational opportunities and social isolation. Comparable individual risk factors, in the form of female gender, social isolation, low-quality information, history of psychological distress, children living in the household, living alone, or single parenthood, have been identified in various reviews [23,24,39]. However, the pandemic regulations did not just create stressors. Preventive measures, such as hygiene measures, protective equipment, and regulated infrastructures in the workplace, had a positive impact on stress, anxiety, and depression [23,25,40,41]. Although the present study showed that private stress increased among employees with children, their PHQ-2 values were significantly lower than among employees without children. Li and Xu also showed that high levels of family support were associated with positive attitudes towards social distancing, mental health, and loneliness [42]. Other preventive factors identified were male gender, higher age group, and working predominantly from home.

### 4.3. Patient-Related Professions

Present results showed a significantly higher burden on patient-related professions during the pandemic. On the one hand, dissatisfaction with the work situation under pandemic conditions, challenges at work, and work-related stress increased significantly compared to other professional groups. On the other hand, partly significantly higher prevalences and intensities for musculoskeletal complaints and a decrease in mental well-being were found for patient-related professions.

El-Hage et al. examined studies from previous pandemics, as well as previously published studies on the COVID-19 pandemic, to identify the following stressors. Organizational stressors include inadequate protective equipment, implementation of rapidly changing regulations, and lack of medications or technical equipment, such as ventilators. Social stressors form concerns about one’s own health or the health of family members, the risk of infection, social stigma, and associated social isolation [43]. The literature review by Mulfinger et al. also showed an increased workload for healthcare workers during pandemics [44]. In addition, Aydin and Atic also concluded that the uncertainty, delay in preventive measures, and rapid transmission of infection associated with the pandemic has led to an increase in stress, depression, and burnout in patient-related professions [45].

In our study, 93.1% of the patient-related professions were affected by (musculoskeletal) complaints. A comparable prevalence among healthcare workers was surveyed by Arca et al. [46]. Musculoskeletal complaints were dominated by shoulder and neck pain, back pain, headache, or burning eyes, with a prevalence between 60–70%. In Arca et al., the neck and back region were most commonly affected [46]. In addition, one in three of four participants showed feelings of general fatigue, and more than half reported sleeping problems. This distress was reflected in the PHQ-4, which showed mild-to-severe psychological distress in 58.3% of patient-related professions. The Umbrella Review by Fernandez et al. and Sahebi et al. on anxiety and depression of healthcare workers confirmed high scores [47,48]. Overall, a decreased quality of life was reported by more than three quarters of the patient-related professions. Almhdawi et al. identified sleep quality, mental health, musculoskeletal health status, and financial status as potential influencing factors in relation to decreased quality of life among nurses during the COVID-19 pandemic [49]. In our study, more than one-third of patient-related professions were taking medications for musculoskeletal complaints. In this context, medication use increased by 64.6% compared to the time before the pandemic.

### 4.4. Limitations

This study shows some limitations that should be considered.

On the one hand, the meaning of the results of the retrospective study design is lower than in prospective or randomized studies. Due to the unpredictability of the development of the pandemic, only two time points were collected. Data collection of the time point before the pandemic allowed for a high rate of up to 68 missing or no data (4%). Moreover, these are based on subjective data given by the employees. On the other hand, the dynamic changes in COVID-19 prescriptions within the survey period of six months was not considered as a confounder.

In addition, the questionnaire was completed by the participants themselves. Response bias, in the sense of tending toward the middle or social desirability, cannot be excluded. It should also be noted that the results of the questionnaires only indicate an increased symptom burden, but do not allow any diagnoses. We did not record whether the employees themselves had COVID-19 or whether they had a post-COVID-19 condition. Musculoskeletal complaints, depression, anxiety, and sleep disorders are the most common symptoms of the post-COVID-19 condition [50,51,52]. In our study, we did not further investigate the correlation of complaints and a post-covid condition.

Nevertheless, the study includes a sample size of 1678 employees. This large cohort allowed us to perform extensive subgroup analyses to identify preventive factors or groups at particular risk. In addition, the employees were recruited from very hereditary sectors, which could indicate that the results can be related to the general population.

## 5. Conclusions

In conclusion, an increase in musculoskeletal disorders and deterioration in the mental well-being of employees in public services during the COVID-19 pandemic was observed. In comparison, patient-related professions were significantly more affected than other occupational groups. Male gender, a younger age group, employees without children, and pandemic-related predominant working from home were identified as preventive factors. With respect to mental health, the preventive factors identified were male gender, a middle- or older-age group, employees with children, and predominantly working from home due to the pandemic.

Attention should be drawn to this development and prevention strategies should be brought into focus to strengthen the employees’ health. Special focus should be drawn to healthcare workers who were particularly confronted with pandemic-associated challenges.

## Figures and Tables

**Table 1 jpm-13-01478-t001:** Socio-demographic data, in total, of patient-related professions (PRP) and other professions (OP).

	Total	PRP	OP
	*n* = 1678 (%)	*n* = 204 (%)	*n* = 1474 (%)
Institution			
University hospital	642 (38.3)	204 (100)	438 (29.7)
Regional council	389 (23.2)		389 (26.4)
District administration	289 (17.2)		289 (19.6)
Relief organization	42 (2.5)		42 (2.8)
Energy supplying company	316 (18.8)		316 (18.8)
Gender			
Female	1045 (62.3)	145 (72.1)	900 (61.1)
Male	617 (36.8)	57 (27.9)	560 (38.0)
Diverse	4 (0.2)		4 (0.3)
Not specified	12 (0.7)		12 (0.8)
Age group			
18–29 years	285 (17.0)	49 (24.0)	236 (16.0)
30–44 years	587 (35.0)	71 (34.8)	516 (35.0)
45–64 years	795 (47.4)	83 (40.7)	712 (48.3)
65+ years	5 (0.3)	1 (0.5)	4 (0.3)
Not specified	6 (0.4)		6 (0.4)
Profession			
Medical service	31 (1.8)	31 (15.2)	
Nurse	173 (10.3)	173 (84.8)	
Administration or pure office work	965 (57.5)		965 (65.5)
Other professions	509 (30.3)		509 (34.5)

**Table 2 jpm-13-01478-t002:** Work situation during the COVID-19 pandemic, in total, of patient-related professions (PRP) and other professions (OP).

	Total	PRP	OP	*p*-Value
	*n* = 1678 (%)	*n* = 204 (%)	*n* = 1474 (%)	
Work location				<0.001
Usual office space	873 (52.0)	195 (95.6)	678 (46.0)	
Usual home office space	32 (1.9)		32 (2.2)	
Pandemic-related as much as possible in working from home	628 (37.4)	5 (2.5)	623 (42.3)	
Pandemic-related only working from home	121 (7.2)	1 (0.5)	120 (8.1)	
Different office space	23 (1.4)	3 (1.5)	20 (1.4)	
Not specified	1 (0.1)		1 (0.1)	
Satisfaction with the work situation due to pandemic				<0.001
Much more satisfied	296 (17.6)	7 (3.4)	289 (19.6)	
Satisfied	899 (53.6)	77 (37.7)	822 (55.8)	
Less satisfied	368 (21.9)	89 (43.6)	279 (18.9)	
Not at all satisfied	107 (6.4)	31 (15.2)	76 (5.2)	
Not specified	8 (0.5)		8 (0.5)	
Facing more challenges				<0.001
Much more	363 (21.6)	97 (47.5)	266 (18.0)	
More	634 (37.8)	80 (39.2)	554 (37.6)	
Remained the same	532 (31.7)	24 (11.8)	508 (34.4)	
Decreased	83 (4.9)	3 (1.5)	80 (5.4)	
Decreased significantly	29 (1.7)		29 (2)	
Not specified	37 (2.2)		37 (2.5)	
Work-related stress				<0.001
Much more	249 (14.8)	71 (34.8)	178 (12.1)	
More	525 (31.3)	80 (39.2)	445 (30.2)	
Remained the same	722 (43.0)	47 (23.0)	675 (45.8)	
Decreased	132 (7.9)	6 (2.9)	126 (8.5)	
Decreased significantly	28 (1.7)		28 (1.9)	
Not specified	22 (1.3)		22 (1.5)	
Private stress				<0.001
Much more	266 (15.9)	53 (26.0%)	213 (14.5%)	
More	526 (31.3)	77 (37.7%)	449 (30.5%)	
Remained the same	587 (35.0)	54 (26.5%)	533 (36.2%)	
Decreased	224 (13.3)	14 (6.9%)	210 (14.2%)	
Decreased significantly	57 (3.4)	6 (2.9%)	51 (3.5%)	
Not specified	18 (1.1)		18 (1.2%)	
Frequency of social contacts				<0.001
Much more	13 (0.8)		13 (0.9)	
Remained the same	124 (7.4)	5 (2.5)	119 (8.1)	
Decreased	528 (31.5)	46 (22.5)	482 (32.7)	
Decreased significantly	1003 (59.8)	151 (74)	852 (57.8)	
Not specified	10 (0.6)	2 (1.0)	8 (0.5)	

**Table 3 jpm-13-01478-t003:** Prevalence of the complaints in the last two weeks, in total, of patient-related professions (PRP) and other professions (OP). Percentage of intensity, frequency, and medication use refer to the affected employees.

	Total	PRP	OP	*p*-Value
Area of Musculoskeletal Complaints or Symptoms	*n* = 1678 (%)	*n* = 204 (%)	*n* = 1474 (%)	
Shoulder and neck				
Prevalence	1042 (62.1)	141 (69.1)	901 (61.1)	0.027
Intensity increased	786 (75.4)	113 (80.1)	673 (74.7)	0.011
Frequency > 7/14 days	487 (46.7)	78 (55.3)	409 (45.4)	0.060
Medication use	187 (17.9)	48 (34.0)	139 (15.4)	<0.001
Back				
Prevalence	890 (53.0)	124 (60.8)	766 (52.0)	0.015
Intensity increased	663 (74.5)	102 (82.3)	561 (73.2)	0.026
Frequency > 7/14 days	361 (40.6)	55 (44.4)	306 (39.9)	0.615
Medication use	159 (17.9)	42 (33.9)	117 (15.3)	<0.001
Lower extremity				
Prevalence	339 (20.2)	59 (28.9)	280 (19.0)	0.001
Intensity increased	269 (79.4)	52 (88.1)	217 (77.5)	0.107
Frequency > 7/14 days	169 (49.9)	32 (54.2)	137 (48.9)	0.336
Medication use	60 (17.7)	15 (25.4)	45 (16.1)	0.229
Headache or burning eyes				
Prevalence	990 (59.0)	143 (70.1)	847 (57.5)	<0.001
Intensity increased	680 (68.9)	109 (76.2)	571 (67.4)	0.039
Frequency > 7/14 days	276 (27.9)	47 (32.9)	229 (27.0)	0.080
Medication use	252 (25.5)	60 (42.0)	192 (22.7)	<0.001
Feeling of general fatigue				
Prevalence	1118 (66.6)	153 (75.0)	965 (65.5)	0.012
Intensity increased	884 (79.1)	137 (89.5)	747 (77.4)	<0.001
Frequency > 7/14 days	468 (41.9)	77 (50.3)	391 (40.5)	<0.001
Medication use	56 (5.0)	6 (3.9)	50 (5.2)	0.798
Sleeping problems				
Prevalence	722 (43.0)	108 (52.9)	614 (41.7)	0.003
Intensity increased	528 (73.1)	85 (78.7)	443 (72.1)	0.085
Frequency > 7/14 days	265 (36.7)	54 (50.0)	211 (34.4)	0.015
Medication use	52 (7.2)	9 (8.3)	43 (7.0)	0.627

**Table 4 jpm-13-01478-t004:** Psychological distress, health status, and quality of life, in total, of patient-related professions (PRP) and other professions (OP).

	Total	PRP	OP	*p*-Value
	*n* = 1678 (%)	*n* = 204 (%)	*n* = 1474 (%)	
PHQ-4				0.048
Normal distress	829 (49.4)	85 (41.7)	744 (50.5)	
Mild distress	594 (35.4)	77 (37.7)	517 (35.1)	
Moderate distress	169 (10.1)	25 (12.3)	144 (9.8)	
Severe distress	60 (3.6)	13 (6.4)	47 (3.2)	
GAD-2				0.136
≥3 points (conspicuous)	258 (15.4)	41 (20.1)	217 (14.7)	
<3 points (inconspicuous)	1401 (83.5)	161 (78.9)	1240 (84.1)	
PHQ-2				0.016
≥3 points (conspicuous)	356 (21.2)	59 (28.9)	297 (20.1)	
<3 points (inconspicuous)	1306 (77.8)	143 (70.1)	1163 (78.9)	
Health status				0.004
Improved	213 (12.6)	22 (10.8)	191 (13.0)	
Remained the same	585 (34.9)	63 (30.9)	522 (35.4)	
Worsened	864 (51.5)	108 (57.8)	756 (51.3)	
Quality of life				<0.001
Improved	230 (13.7)	22 (10.8)	208 (14.1)	
Remained the same	264 (15.7)	24 (11.8)	240 (16.3)	
Worsened	1153 (68.7)	157 (77.0)	996 (67.6)	

## Data Availability

The data presented in this study are available on request from the corresponding author.

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
