# Peer review of "Influence of the COVID-19 Pandemic on Musculoskeletal Complaints and Psychological Well-Being of Employees in Public Services—A Cohort Study"

_jpm, 2023, doi:10.3390/jpm13101478_

Round 1
Reviewer 1 Report
The authors used an online questionnaire to assess pre- to post-Sars-COV19 pandemic changes in musculoskeletal pain, mental well-being, health status and quality of life, mostly in patient-related professionals. They found an increase in musculoskeletal disorders and a decrease in psychological well-being. The paper adds further evidence about the impact of Sars-COV19 in Germany. The issue has been investigated in the general population and specific patients groups including the so called post-COVID19 syndrome (Aydin and Atiç, 2023; Poyurovsky and Weizman, 2023; Talhari et al., 2023; Enax-Krumova et al, 2023; Almhdawi et al, 2023). The authors should discuss other works related to their findings.
Were patients screened for comorbidities worsened by SarsCOV2?
Table 1, line 11 from top: Female 1045 (62.3) 145 (72.1) 1000 (61.1) is not correct.
Table 2. Fully and essentially should be probably replaced by more and much more; the same hold true for the items of PHQ-2 and PHQ-4. Also, you should be more consistent. Less should diminished.
Table 4. Conspicuous and incospiscuous should be replaced by the level of anxiety (ie, minimal, milde etc).
The paper could be of interest because of the social and medical organization of Germany. In any event, the authors should be more consistent in terminology and improve english. Perhaps the paper should be by cut by 30-50%.
Colloquial terms and technical terminology should be improved, I guess
Author Response
Dear Reviewer,
thank you very much for your faithful review and further input. Subsequent we will respond point-by-point to your comments. We tried to do our best to revise our manuscript accordingly.
Kind regards
“The authors used an online questionnaire to assess pre- to post-Sars-COV19 pandemic changes in musculoskeletal pain, mental well-being, health status and quality of life, mostly in patient-related professionals. They found an increase in musculoskeletal disorders and a decrease in psychological well-being. The paper adds further evidence about the impact of Sars-COV19 in Germany. The issue has been investigated in the general population and specific patients groups including the so called post-COVID19 syndrome (Aydin and Atiç, 2023; Poyurovsky and Weizman, 2023; Talhari et al., 2023; Enax-Krumova et al, 2023; Almhdawi et al, 2023). The authors should discuss other works related to their findings.
Thank you very much for your valuable input. In our study, we did not survey whether the employees themselves had SarsCOV2 or were affected by the “post-COVID-19 syndrome". We cited the mentioned studies in our manuscript and discussed them in the context of our findings.
Were patients screened for comorbidities worsened by SarsCOV2?”
We did not screen participants for comorbidities. Using a self-reported questionnaire in our study, we were less concerned about a diagnosis than about a change in complaints in the context of the pandemic and possible influencing factors.
Table 1, line 11 from top: Female 1045 (62.3) 145 (72.1) 1000 (61.1) is not correct.
There was a number error there. The correct number is 900 (61.1) and was changed accordingly (line 11 from top).
“Table 2. Fully and essentially should be probably replaced by more and much more; the same hold true for the items of PHQ-2 and PHQ-4. Also, you should be more consistent. Less should diminished.”
Thank you for the hint. We replaced the terms accordingly. In addition, corona pandemic was uniformly changed to COVID-19 pandemic. Musculoskeletal disorders were uniformly changed to musculoskeletal complaints.
„Table 4. Conspicuous and inconspiscuous should be replaced by the level of anxiety (ie, minimal, milde etc).”
Thank you for your hint. We changed the terms accordingly.
“The paper could be of interest because of the social and medical organization of Germany. In any event, the authors should be more consistent in terminology and improve english. Perhaps the paper should be by cut by 30-50%.“
Thank you for your thoroughful review. We changed the terms accordingly to be more consistent in terminology and shortened the manuscript. Also, the manuscript was proofread by a native speaker, again.
Please see the attachment for the revisioned version.

Reviewer 2 Report
Dear Authors:
First of all I would like to congratulate you for the work done, as it can be seen that it is a mnuscript of quality and scientific consistency. Having said this, I must tell you that the subject and the objective of your study is well known by the scientific community, since we are talking about dysfunctions at the physical and psychological level during the coronavirus pandemic. Fortunately, the pandemic and the restrictions have gone down in history and it is already known by all the after-effects they left in the past, in addition to the effects on the organism. I have nothing against the manuscript, it is very good and I do not see any flaws, but it lacks interest for the reader, since there are more articles with the same subject and study design.
Regards.
Author Response
Dear Reviewer,
Thank you very much for the honest and critical discussion of our manuscript. In the meantime numerous studies evaluated the mental and physical health of healthcare workers during the pandemic. The aim of our study was to point out differences between the occupational groups in the public sector and possible protective factors. We aimed to investigate the interplay of physical and mental health against the background of changes in the professional setting as well as in private life due to the Sars-Cov19 pandemic. Preventive factors were identified for employees in the public sector, that should receive more attention in light of impending restrictions due to again increasing incidences. We believe, that with our manuscript we can contribute valuable data from the public sector and especially from health care workers to the community. Against the background of shortage of skilled workers as well as the many challenges facing our healthcare systems, it seems important to us to identify the concerns of the healthcare sector. We would therefore be very pleased to receive a positive revision. Please see the attachment for the revisioned version.
Kind regards

Round 2
Reviewer 1 Report
Terminology should be more consistent and further improved.
The authors put some effort to improve the paper. In my opinion, english should be further improved. Given the existing literature on the subject, the paper should probably be shortened by 30-50%. I leave to the Editor the final decision.
Reviewer 2 Report
Dear authors:
Thank you for your clarifications and for making such enlightening comments to substantiate your research. At first I did not see any relevant contribution, but with the clarification of the subject matter, I welcome your article as the design and methodology is correct.
Best regards.